# Fluid flow channeling and mass transport with discontinuous porosity distribution

Simon Boisserée<sup>1</sup>, Evangelos Moulas<sup>2</sup>, and Markus Bachmayr<sup>1</sup>

<sup>1</sup>Institut für Geometrie und Praktische Mathematik, RWTH Aachen University, Templergraben 55, 52056 Aachen, Germany

<sup>2</sup>Institute of Geosciences and Mainz Institute of Multiscale Modeling, Johannes Gutenberg-Universität Mainz,

J.-J.-Becher-Weg 21, 55128 Mainz, Germany

**Correspondence:** Simon Boisserée (boisseree@igpm.rwth-aachen.de)

Abstract. The flow of fluids within porous rocks is an important process with numerous applications in Earth sciences. Modeling the compaction-driven fluid flow requires the solution of coupled nonlinear partial differential equations that account for the fluid flow and the solid deformation within the porous medium. Despite the nonlinear relation of porosity and permeability that is commonly encountered, natural data show evidence of channelized fluid flow in rocks that have an overall layered structure. Layers of different rock types have discontinuous hydraulic and mechanical properties. We present numerical results obtained by a novel space-time method, which can handle discontinuous initial porosity (and permeability) distributions efficiently. The space-time method enables straightforward coupling to models of mass transport for trace elements. Our results indicate that, under certain conditions, the discontinuity of the initial porosity influences the distribution of incompatible trace elements, leading to sharp concentration gradients and large degrees of elemental enrichment. Finally, our results indicate that the enrichment of trace elements depends not only on the channelization of the flow but also on the interaction of fluid-filled channels with layers of different porosity and permeability.

# 1 Introduction

The flow of fluids in the Earth's subsurface is important for many applications. Examples of such applications include, but are not limited to, the migration of magma McKenzie (1984); Barcilon and Richter (1986), the flow of glaciers Fowler (1984), the integrity of subsurface reservoirs Räss et al. (2018); Yarushina et al. (2022), and the efficiency of geothermal systems Utkin and Afanasyev (2021). A distinctive aspect of the fluid flow within the deep Earth is that rocks cannot be treated as purely elastic or rigid, requiring consideration of their bulk (volumetric) viscous deformation McKenzie (1984); Scott and Stevenson (1986). In fact, recent experiments have confirmed that the viscous/viscoelastic behavior of rocks can be observed also at near-surface conditions Sabitova et al. (2021). Thus, the volumetric deformation and the associated fluid flow need to be considered in a coupled fashion since (de)compaction can drive fluid flow and vice versa Connolly and Podladchikov (1998); Vasilyev et al. (1998). In the latter studies, *porosity waves* were observed numerically. Such waves reflect the propagation of porosity perturbations (and the associated volumetric deformation) in a wave-like fashion with minimal dissipation. The transport of fluid-filled porosity in a non-dissipative fashion has been at the focus of research by geoscientists since it has

important implications for the geochemical anomalies that are observed near the surface of the Earth Richter (1986); Navon and Stolper (1987); Jordan et al. (2018).

The shape of porosity waves has been shown to depend very sensitively on the nonlinear behavior of the bulk (volumetric) viscosity. For example, in cases where the compaction/decompaction behavior is associated with significant changes in the effective viscosity, porosity waves take a channel-like shape (in two or three dimensions) that is responsible for the focusing of the flow towards the Earth's surface Räss et al. (2018); Connolly and Podladchikov (2007); Räss et al. (2014, 2019); Yarushina and Podladchikov (2015); Yarushina et al. (2015, 2020). The focusing of the flow produces "chimney-like" features that resemble geophysical observations Räss et al. (2018); Yarushina et al. (2020). The occurrence of such features is very important in the quantification of fluid flow and the associated geochemical anomalies Spiegelman and Kelemen (2003).

An essential feature of geological formations is that rocks are typically found in layers (strata). The layers are often composed of rock types that have different physical properties, such as porosity and permeability. It is exactly this change in permeability that is responsible for the formation of geological reservoirs. For example, a typical underground reservoir must be composed of rocks of high porosity (and permeability) and must be covered by rocks of negligible porosity (and permeability) that act as a "seal" to the underlying rock units. This configuration typically requires the consideration of porosity (and permeability) jump discontinuities across the lithological boundaries. However, the methods used to solve the respective poro-viscoelastic equations numerically cannot handle a discontinuous initial porosity, and hence only approximate it by a continuous function with steep gradient. This approach leads to smoothing effects and does not preserve the discontinuous nature of solutions. Resolving the solution behavior next to a discontinuity is crucial in all the applications where the quantification of the fluid flow is needed and can thus be important for safety analyses in geoengineering applications Yarushina et al. (2022).

Here, we consider a poro-viscoelastic model that generalizes the one introduced in Connolly and Podladchikov (1998); Vasilyev et al. (1998) for the interaction of porosity and pressure. For modeling sharp transitions between materials, as caused, for example, by stacked rock layers, it is important to be able to treat porosities with *jump discontinuities*. These discontinuities turn out to be determined mainly by the initial condition, as it was shown in Bachmayr et al. (2023) based on results from Simpson et al. (2006) for smooth initial porosities. In addition, we utilize a newly-developed space-time method that has been shown to be more accurate in solving this particular problem in the presence of jump discontinuities Bachmayr and Boisserée (2025). Our approach can be used to benchmark methods that do not include discontinuities and quantify the error between the two approaches. An additional advantage of the space-time method is that it can be coupled to simple models of chemical-tracer transport (see, for example, Richter, 1986; Jordan et al., 2018) as a post-processing step, since the entire porosity-pressure history is saved and the chemical transport problem does not feedback into the porosity-pressure (hydro-mechanical) model. The results obtained from this coupling allow us, for the first time, to investigate the evolution of chemical anomalies in the presence of channelized fluid flow, and their interactions with porosity/permeability discontinuities. In particular, our results are relevant to the formation of ore deposits and to the transport of trace elements in the subsurface.

# 1.1 The governing equations

The model for poro-viscoelastic flow on which we focus in this work reads

$$\partial_t \phi = -(1 - \phi) \left( \phi^m \frac{p}{\eta_b \sigma(p)} + Q \partial_t p \right), \qquad \phi(0, \cdot) = \phi_0, \qquad (1a)$$

$$\partial_t p = \frac{1}{Q} \left( \operatorname{div}_x \left( \frac{k_b}{\mu \phi_b^n} \phi^n (\nabla_x p + (1 - \phi) \delta \rho g e_d) \right) - \phi^m \frac{p}{\eta_b \sigma(p)} \right), \qquad p(0, \cdot) = p_0, \tag{1b}$$

as previously described in Connolly and Podladchikov (1998); Vasilyev et al. (1998). Here,  $\phi$  denotes the porosity (void ratio), p is the effective pressure,  $\sigma$  accounts for *decompaction weakening* Räss et al. (2018, 2019), Q is the compressibility (equal to  $K^{-1}$ , where K is the bulk modulus), and  $\delta \rho = \rho^s - \rho^f$  the density difference. Furthermore,  $\nabla_x f = (\partial_{x_1} f, \dots, \partial_{x_d} f)^{\top}$  and  $\operatorname{div}_x \mathbf{f} = \sum_{i=1}^d \partial_{x_i} f_i$  for functions  $f : \mathbb{R}^d \to \mathbb{R}$ ,  $\mathbf{f} : \mathbb{R}^d \to \mathbb{R}^d$  as usual. For any function g(t,x), we denote  $g(0,\cdot)$  as the function g at a fixed time t=0 with varying x. Finally, t is time and  $e_d$  is the vector indicating the gravity acceleration direction (all symbols and the respective units are given in Table 1). The problem is furthermore supplemented with initial porosity  $\phi_0 : \Omega \to (0,1)$  and initial effective pressure  $p_0 : \Omega \to \mathbb{R}$ .

| Symbol                    | Meaning                                                                           | Unit                              | Value                                          |
|---------------------------|-----------------------------------------------------------------------------------|-----------------------------------|------------------------------------------------|
| $\phi$                    | porosity                                                                          |                                   |                                                |
| $\phi_{ m b}$             | background porosity                                                               |                                   | $10^{-3}$                                      |
| p                         | effective pressure                                                                | Pa                                |                                                |
| $\mathcal C$              | total concentration                                                               | $\mathrm{kg}\cdot\mathrm{m}^{-3}$ |                                                |
| $\eta_{ m b}$             | bulk viscosity                                                                    | $Pa \cdot s$                      | $10^{19}$                                      |
| K                         | bulk modulus                                                                      | Pa                                | $3 \cdot 10^9$                                 |
| $k_{ m b}/\mu$            | permeability over fluid viscosity                                                 | $m^2 \cdot Pa^{-1} \cdot s^{-1}$  | $10^{-17}$                                     |
| n                         | Carman-Kozeny exponent                                                            |                                   | 3                                              |
| m                         | viscosity exponent                                                                |                                   | 2                                              |
| g                         | gravity                                                                           | $\mathrm{m}\cdot\mathrm{s}^{-2}$  | 10                                             |
| $\rho^{\rm f}$            | fluid density                                                                     | $\mathrm{kg}\cdot\mathrm{m}^{-3}$ | 2500                                           |
| $\rho^{\rm s}$            | solid density                                                                     | $\mathrm{kg}\cdot\mathrm{m}^{-3}$ | 3000                                           |
| $\chi^{ m f}$             | fluid mass fraction                                                               |                                   |                                                |
| $\chi^{ m s}$             | solid mass fraction                                                               |                                   |                                                |
| $K_D$                     | concentration ratio $\frac{\rho^{\rm s} \chi^{\rm s}}{\rho^{\rm f} \chi^{\rm f}}$ |                                   | $10^{-3}$                                      |
| $\mathbf{v}^{\mathrm{f}}$ | fluid velocity                                                                    | $\mathrm{m}\cdot\mathrm{s}^{-1}$  |                                                |
| $\mathbf{v}^{\mathrm{s}}$ | solid velocity                                                                    | $\mathrm{m}\cdot\mathrm{s}^{-1}$  |                                                |
| T                         | total time                                                                        | S                                 | $4.73364 \cdot 10^{13} \; (1.5  \mathrm{Myr})$ |

Table 1. Variables and physical quantities

An extension of the hydro-mechanical model (1) is to consider the transport of a chemical tracer (such as a trace element) as described in (Jordan et al., 2018, Sec. 3). In particular, the trace-element transport equations are chosen since we consider

that the abundance of trace elements does not affect the mechanical or the hydraulic properties of the rock. As a consequence, the trace-element transport problem depends on the hydromechanical problem, but the opposite is not true. This allows us to treat the chemical transport problem as a post-processing step after we have calculated the respective fluid velocities and the porosity distribution. The amount of tracer is quantified using the total concentration

$$\mathcal{C} = \phi \rho^{f} \chi^{f} + (1 - \phi) \rho^{s} \chi^{s}, \tag{2}$$

and fulfills the transport equation

75 
$$\partial_t C + \operatorname{div}_x(\mathbf{v}^e C) = 0.$$
 (3)

Here  ${\bf v}^{\rm e}$  denotes the effective velocity and at the limit where  ${\bf v}^{\rm s}\approx 0$  holds, is given by

$$\mathbf{v}^{e} = \frac{\mathbf{v}^{f} \phi}{\phi + (1 - \phi) K_{D}}, \qquad \mathbf{v}^{f} = \frac{1}{\phi} \frac{k_{b}}{\mu \phi_{b}^{n}} \phi^{n} \left( \nabla_{x} p + (1 - \phi) \delta \rho g \mathbf{e}_{d} \right), \tag{4}$$

where  $K_D = \frac{\rho^s \chi^s}{\rho^f \chi^f}$  describes the concentration ratio of the tracer which is assumed to be constant as already indicated in Table 1. Note that, in this case,  $K_D$  is a ratio of concentrations and not of mass fractions. Furthermore, equation (3) assumes that porosity is continuous and its derivation can be found in Appendix A. For cases where porosity is discontinuous, the jump condition must guarantee the conservation of mass at the discontinuity (see Appendix B for details).

#### 1.2 Applicability of assumptions

90

The previous hydro-mechanical system of equations (1) results from the simplification of the multiphase-, viscoelastic-Stokes' equations at the static limit. The static limit occurs when no far-field stresses are imposed at the boundaries, and the buoyancy stresses are relatively small within the model domain. This limit is justified in cases where the effective pressure is close to zero. In such cases, the shear stresses that rocks can support are very small and, in many applications, can be assumed to be negligible Aharonov et al. (1997); Connolly and Podladchikov (1998, 2007); Scott and Stevenson (1984). Being close to the static limit implies that the solid velocity for the mechanical problem is taken at the limit where  $\mathbf{v}^f \gg \mathbf{v}^s \approx 0$  (but generally  $\operatorname{div}_x \mathbf{v}^s \neq 0$ ).

One particular process where the trace-element transport is important is when melt is ascending within the Earth's mantle Richter (1986). In regions such as in the mantle wedge or within a mantle plume, the temperature does not change significantly. In these geodynamic environments, the confining pressure is large and the melt-filled porosity of the mantle rock is very small, typically in the order of 0.001-0.01 Sims et al. (1999). To model the trace-element equilibrium and the chemical interaction between solid and fluid, we use the partition coefficient  $K_D$ . The partition coefficient changes as a function of the mineralogy of the rock, its pressure and its temperature. However, for a given material, the variation of the partition coefficient with pressure is very weak and can be considered constant over several GPa of pressure Taura et al. (1998).

Having  $\mathbf{v}^f$  from (1) allows the solution of (3). The specific form of (3) is valid at the limit where grain-scale chemical diffusion and hydro-dynamic dispersion are ignored. Previous studies indicate that, on the large scales considered here, these phenomena can be neglected Richter (1986); Stavropoulou et al. (1998).

# 100 1.3 Existing numerical methods

Various methods have been proposed to solve the hydro-mechanical problem (1) numerically, for example finite difference schemes with implicit time-stepping in Connolly and Podladchikov (1998) and adaptive wavelets in Vasilyev et al. (1998). In a number of recent works, pseudo-transient schemes based on explicit time stepping in a pseudo-time variable have been investigated. Due to their compact stencils, low communication overhead and simple implementation, such schemes are well suited for parallel computing on GPUs, so that very high grid resolutions can be achieved to compensate the low order of convergence, as shown for example in Räss et al. (2018); Utkin and Afanasyev (2021); Räss et al. (2014, 2019); Yarushina et al. (2020); Reuber et al. (2020). Even though all of these schemes are observed to work well for smooth initial porosities  $\phi_0$ , their convergence can be very slow in problems with non-smooth  $\phi_0$ , in particular in the presence of discontinuities Bachmayr and Boisserée (2025). Examples of this behavior are also shown in Appendix C. In such cases, due to the smoothing that is implicit in the finite difference schemes, accurately resolving sharp localized features can require extremely large grids that are computationally inefficient.

## 1.4 Novel contributions

Our approach considers the utilization of a space-time method to solve the hydro-mechanical problem (1). This method was introduced for this particular problem in Bachmayr and Boisserée (2025), and has the advantage that the entire solutions of porosity and effective pressure fields can be stored in a space-time grid. In addition, this approach can handle discontinuities in the porosity  $\phi$  without approximating it by a continuous function with steep gradient. As a result, smaller grids and less computational effort are needed compared to continuous schemes such as finite differences. Since the method generates efficient approximations of the entire time history of a solution to (1) in a sparse format, its coupling to the problem of chemical-tracer transport (CT) given by (3) becomes straightforward. This is because the CT problem does not give feedback to the model (1), and thus solving it can be seen as post-processing step.

## 1.5 Outline

Since our results from the HM model (1) are uncoupled to the results of the CT problem (3), we begin with a short description of the methods used to solve the HM model. In Section 2 we introduce the methods to obtain the numerical results both for the HM model in Section 3 and for the CT problem in Section 4. We finish with a discussion regarding the implication of our results for the porous fluid transport in natural systems.

## 2 Methods

#### 2.1 Hydro-mechanical model (HM)

To solve (1) we consider the space-time adaptive method which was introduced in Bachmayr and Boisserée (2025) based on a combination of a Picard iterations for (1a) and a particular adaptive least squares discretization of (1b) which itself is based on

Führer and Karkulik (2021); Gantner and Stevenson (2021, 2024). To make this more precise, we start by introducing the new variable

$$\varphi = -\log(1 - \phi),\tag{5}$$

so that  $\phi = 1 - e^{-\varphi}$ . The previous transformation allows the investigation of cases where the porosity is larger than the typical "small-porosity limit" Vasilyev et al. (1998). The system (1) can then be written in the form

$$\partial_t \varphi = -\left(\beta(\varphi) \frac{p}{\sigma(p)} + Q \partial_t p\right),$$
 (6a)

$$\partial_t p = \frac{1}{Q} \left( \operatorname{div}_x \left( \alpha(\varphi) (\nabla_x p + \zeta(\varphi)) \right) - \beta(\varphi) \frac{p}{\sigma(p)} \right), \qquad p(0, \cdot) = p_0,$$
(6b)

where

$$\alpha(\varphi) = \frac{k_{\rm b}}{\mu \phi_{\rm b}^n} (1 - e^{-\varphi})^n, \quad \beta(\varphi) = \frac{1}{\eta_{\rm b}} (1 - e^{-\varphi})^m, \quad \zeta(\varphi) = e^{-\varphi} \delta \rho g \boldsymbol{e}_d. \tag{7}$$

To solve (6b) for a fixed  $\overline{\varphi}$  we consider a linearization, that is, we solve

$$\partial_t p^{(k)} = \frac{1}{Q} \left( \operatorname{div}_x \left( \alpha(\overline{\varphi}) \left( \nabla_x p^{(k)} + \zeta(\overline{\varphi}) \right) \right) - \beta(\overline{\varphi}) \frac{p^{(k)}}{\sigma(p^{(k-1)})} \right), \quad p^{(k)}(0, \cdot) = p_0,$$
 (8)

for  $p^{(k)}$  given the previous iterate  $p^{(k-1)}$ . By defining

$$G[p^{(k-1)}](p^{(k)}, \psi^{(k)}) = \begin{pmatrix} \frac{1}{Q} \left( \operatorname{div}(p^{(k)}, \psi^{(k)}) + \beta(\overline{\varphi}) \frac{p^{(k)}}{\sigma(p^{(k-1)})} \right) \\ \psi^{(k)} + \alpha(\overline{\varphi}) \nabla_x p^{(k)} \\ p^{(k)}(0, \cdot) \end{pmatrix}, \quad R = \begin{pmatrix} 0 \\ -\alpha(\overline{\varphi}) \zeta(\overline{\varphi}) \\ p_0 \end{pmatrix}, \tag{9}$$

with  $\operatorname{div}(p,\psi)=\partial_t p+\operatorname{div}_x\psi$ , we can reformulate (8) as first-order system

$$G[p^{(k-1)}](p^{(k)}, \psi^{(k)}) = R.$$
 (10)

Note that the second row of (9) can be rewritten as  $\psi^{(k)} = -\alpha(\overline{\varphi})(\nabla_x p^{(k)} + \zeta(\overline{\varphi}))$ . Plugging  $\psi^{(k)}$  into  $\operatorname{div}(p^{(k)}, \psi^{(k)})$  in the first row of (9) yields (8).

Numerically we now use the approach presented in Führer and Karkulik (2021); Gantner and Stevenson (2021, 2024), and thus calculate a least squares minimizer with respect to an appropriately chosen norm. Using the numerical approximation of  $p[\overline{\varphi}]$  from (10) we solve (6a) by discretizing the iteration

$$\varphi^{(k+1)}(t,\cdot) = Q(p_0 - p[\varphi^{(k)}](t,\cdot)) - \int_0^t \beta(\varphi^{(k)}) \frac{p[\varphi^{(k)}](s,\cdot)}{\sigma(p[\varphi^{(k)}](s,\cdot))} ds.$$
 (11)

which is based on integrating (6a) in time. For more details including proofs of convergence we refer to (Bachmayr and Boisserée, 2025, Sec. 3, 4).

Figure 1. Example of a porosity channel at  $T = 1.5 \,\mathrm{Myr}$  (a) with the associated adaptive space-time grid (b); the color of each grid cell denotes its refinement level

This scheme can generate space-time grids corresponding to spatially adapted time steps; an example of this can be found in Figure 1. Furthermore, the method provides computable *a-posteriori* estimates of the error with respect to the exact solution of the coupled nonlinear system (6). Therefore, this method can be used to steer an adaptive grid refinement routine which yields efficient approximations of localized features of solutions, in particular in the presence of discontinuities. In addition, one obtains optimal convergence rates for  $\phi$  and p independent of the presence of discontinuities in  $\phi$ , as observed in (Bachmayr and Boisserée, 2025, Sec. 5.2).

## 2.2 Chemical-tracer transport model (CT)

To solve the chemical transport equation (3), we follow its characteristics. Namely, we consider

$$\partial_t x(t) = \mathbf{v}^{\mathbf{e}}(t, x(t)),$$

$$c(t) = \mathcal{C}(t, x(t)).$$
(12)

Then we calculate

$$\partial_{t}c(t) = \partial_{t}\mathcal{C}(t, x(t)) + \nabla_{x}\mathcal{C}(t, x(t)) \cdot \partial_{t}x(t)$$

$$= -\operatorname{div}_{x}(\mathbf{v}^{e}(t, x(t))\mathcal{C}(t, x(t))) + \nabla_{x}\mathcal{C}(t, x(t)) \cdot \mathbf{v}^{e}(t, x(t))$$

$$= -\operatorname{div}_{x}(\mathbf{v}^{e}(t, x(t)))\mathcal{C}(t, x(t))$$

$$= -\operatorname{div}_{x}(\mathbf{v}^{e}(t, x(t)))\mathcal{C}(t), \qquad (13)$$

the previous yields a coupled system of ordinary differential equations (ODEs)

$$\partial_t x(t) = \mathbf{v}^{\mathbf{e}}(t, x(t)),$$

$$\partial_t c(t) = -\operatorname{div}_x(\mathbf{v}^{\mathbf{e}}(t, x(t))) c(t),$$
(14)

that is used to solve (3). The solution is provided along the characteristics given by  $\mathbf{v}^e$  starting with some initial value  $x_0$  and initial concentration  $c_0$ . We use an explicit Euler scheme to solve (14) for many different starting values  $x_0$ . Note that this approach is highly parallelizable, since we need to solve a high number of independent ODEs for each starting value. By exploiting this we usually achieve very low wall-clock times, even for many starting values corresponding to a high resolution. Note furthermore that this approach only conserves the quantity  $\mathcal{C}$  if  $\mathbf{v}^e$  is continuous. For the discontinuous cases one needs to ensure continuity of the flux and we refer to Section B for more details.

#### 175 **2.3** Model parameters

The model parameters can be derived by non-dimensionalizing the physical models (1) and (3) with values given in Table 1. Choosing the independent scales

$$x^{\text{sc}} = 10^4 \,\text{m}, \qquad \delta \rho^{\text{sc}} g^{\text{sc}} = 5 \cdot 10^3 \,\text{kg} \cdot \text{m}^{-2} \cdot \text{s}^{-2}, \qquad \eta_b^{\text{sc}} = 10^{19} \,\text{Pa} \cdot \text{s},$$
 (15)

yields the dependent scales

80 
$$p^{\text{sc}} = \delta \rho^{\text{sc}} g^{\text{sc}} x^{\text{sc}} = 5 \cdot 10^7 \,\text{Pa},$$
  
 $t^{\text{sc}} = \frac{\eta_{\text{b}}^{\text{sc}}}{p^{\text{sc}}} = 2 \cdot 10^{11} \,\text{s},$   
 $\frac{k_{\text{b}}^{\text{sc}}}{\mu^{\text{sc}}} = \frac{(x^{\text{sc}})^2}{\eta_{\text{b}}^{\text{sc}}} = 10^{-11} \,\text{m}^2 \cdot \text{Pa}^{-1} \cdot \text{s}^{-1}.$  (16)

Hence we end up with the nondimensional parameters

$$\tilde{\eta}_{b} = 1, \quad \frac{\tilde{k}_{b}}{\tilde{\mu}\phi_{b}^{n}} = 1000, \quad \delta\tilde{\rho}\tilde{g}\boldsymbol{e}_{d} = \begin{pmatrix} 0\\1 \end{pmatrix}, \quad B\tilde{Q} = Q\delta\rho^{\mathrm{sc}}g^{\mathrm{sc}}x^{\mathrm{sc}} = \frac{1}{60},$$
(17)

where  $B = Q^{\rm sc} \delta \rho^{\rm sc} g^{\rm sc} x^{\rm sc}$  denotes a non-dimensional number that is the ratio of buoyancy stress to bulk modulus. Note that this may look similar to the Deborah number defined in Connolly and Podladchikov (1998), however, the lengthscale  $x^{\rm sc}$  is taken as an independent quantity in our approach, while in other studies it is derived from the compaction length Connolly and Podladchikov (1998); Vasilyev et al. (1998).

Due to the given length scale and time scale, we can directly translate physical times and domain sizes into our model parameters if we divide by  $x^{\rm sc}$  or  $t^{\rm sc}$ , respectively. For  $T=1.5\,{\rm Myr}$ , this corresponds to

$$\check{T} = T/t^{\text{sc}} = \frac{1.5 \cdot 10^6 \cdot 365.25 \cdot 24 \cdot 60 \cdot 60}{2 \cdot 10^{11}} = 236.682.$$

Note that, in the following, the " $\sim$ " symbols are omitted for convenience. Furthermore, we consider  $\sigma$ , as suggested in Räss et al. (2018, 2019), which is an expression of the form

$$\sigma(v) = 1 - \frac{1 - c_1}{2} \left( 1 + \tanh\left(-\frac{v}{c_2}\right) \right)$$

$$= \frac{c_1 + \exp(2v/c_2)}{1 + \exp(2v/c_2)}, \quad v \in \mathbb{R},$$
(18)

and provides a phenomenological model for decompaction weakening. Here  $c_1 \in (0,1]$  and  $c_2 > 0$ , where  $1 + \tanh$  can be regarded as a smooth approximation of a step function taking values in the interval (0,2). In the most well-studied case  $c_1 = 1$ , as considered in Vasilyev et al. (1998), one observes the formation of porosity waves, whereas the case of  $c_1 

**Figure 2.** Three initial porosity distributions  $\phi_0^a$  (a),  $\phi_0^b$  (b) and  $\phi_0^c$  (c)

We start with the well-known scenario of the smooth initial porosity  $\phi_0^a$  as in Figure 2 (a) and compare the results without decompaction weakening ( $\sigma^b$ ) in Figure 3. The resulting plots show the expected spreading of the fluid front in the case without decompaction weakening in Figure 3 (a,c). In contrast, the fluid flow is focused

Figure 3. Porosity (a,b) and effective pressure (c,d) after  $T=1.5\,\mathrm{Myr}$  for a smooth initial condition ( $\phi_0^\mathrm{a}$ ) without decompaction weakening ( $\sigma^\mathrm{a}$ ) (a,c) and with decompaction weakening ( $\sigma^\mathrm{b}$ ) (b,d)

in the presence of weakening as one can see in Figure 3 (b,d). These results are used as reference and will not be discussed further since they confirm previous findings Räss et al. (2018); Connolly and Podladchikov (1998, 2007); Yarushina et al. (2015).

Figure 4 shows the results of the two initial conditions  $\phi_0^b$  and  $\phi_0^c$  from Figure 2 (b,c) that consider an initial porosity discontinuity. Both results consider the case without decompaction weakening. One can see very sharp transitions of  $\phi$  at the locations of the initial discontinuities. Note that the discontinuity itself cannot move since the model (1) was derived under the assumption that  $\mathbf{v}^s \approx 0$  and porosity is a property of the solid. This agrees with the theoretical results shown in (Bachmayr et al., 2023, Thm. 4.6). As it is especially visible for the porosity distribution, the sign of its transition (positive or negative) depends on whether the initial porosity of the upper layer was smaller (negative jump) or larger (positive jump) compared to the porosity of the underlying layer. Figure 5 shows a cross section of Figure 4 that shows the discontinuities in  $\phi$  more clearly. In contrast, the solution for p is continuous which aligns with the theory derived in (Bachmayr et al., 2023, Sec. 4).

225

Figure 4. Porosity (a,b) and effective pressure (c,d) without decompaction weakening ( $\sigma^a$ ) after  $T=1.5\,\mathrm{Myr}$  for  $\phi^b_0$  (a,c) and  $\phi^c_0$  (b,d)

Figure 6 shows the effect of decompaction weakening on the same initial discontinuous configurations (case  $\sigma^b$ ). In the case of the negative jump ( $\phi_0^b$ ), the channel has a slightly higher maximal porosity compared to the continuous case. Furthermore, Figure 6 (a) shows that there is a very steep increase in porosity at the place of the discontinuity. On the other hand, for the positive jump ( $\phi_0^c$ ), the channel focuses significantly before spreading in the high-porosity/permeability zone as it is shown in Figure 6 (b). This can be expected: we see a narrower channel in the domain that has smaller porosity, but the channel spreads quickly once the fluid enters the domain of high porosity and permeability. This shows that the fluid does not need to channelize as much as in the less porous domain in order to travel upwards.

# 4 Chemical-transport model results

230

An extension of the general model (1) is to consider the transport of a chemical tracer by solving (3). This is achieved by following its characteristics as described in Section 2.2. The natural range of partition coefficients can be very large Irving (1978) and their magnitudes depend on several parameters Karato (2016). However, as it was described in Section 1.2, it is reasonable to assume that the partition coefficient can be approximated as constant given a limited range of temperatures and

Figure 5. Cross section of Figure 4 for  $x_1 = 5 \,\mathrm{km}$  with porosity (a,b) and effective pressure (c,d)

245

250

255

constant mineralogical composition (that is implicitly assumed in our models). Without loss of generality, we examine the case where  $K_D=10^{-3}$  (as already indicated in Table 1) to consider incompatible elements. Incompatible elements are those that partition preferentially in the fluid. Solving for  $\mathcal{C}$  and using the prescribed values for  $\rho^s$ ,  $\rho^f$  and  $K_D$  from Table 1 directly yields  $\chi^s$  and  $\chi^f$  as well.

In this part, we will only plot the normalized chemical tracer  $C/C_0$ . This allows us to quantify the overall enrichment or depletion of a trace element with respect to the initial configuration. Note that since we use a characteristics-based approach for the advection of chemical elements, the chemical evolution is calculated only for the areas where the characteristics are initialized. As a result, the domain where the chemical evolution is calculated, changes in time depending on the effective velocity  $\mathbf{v}^{\mathrm{e}}$ . For simplicity, we consider constant initial data  $C_0(x) = 1$  since C in (3) can be scaled arbitrarily without affecting the solution.

Figure 7 shows the normalized tracer compositions  $\mathcal{C}/\mathcal{C}_0$  connected to the solutions shown in Figure 3 (with  $\phi_0^a$ ) for  $K_D=10^{-3}$ . We see the distribution of an incompatible element that prefers to stay with the fluid, and hence, it gets transported efficiently while draining the area of origin. The role of decompaction weakening becomes more apparent in the case of the channelization of the fluid flow, as shown in Figure 7 (b). In that case, we observe a more pronounced enrichment in the region defined by the fluid-rich channel. It is important to note that this enrichment occurs in both the solid and the fluid, and it occurs at the expense of the trace element's distribution in the source region.

The solution of the CT problem having initial discontinuous porosity  $\phi_0$  is shown in Figure 8. This figure is calculated based on the HM model (porosity-pressure evolution) shown in Figure 4 and assumes no decompaction weakening ( $\sigma^a$ ). The results generally agree with the previous findings that show that the incompatible elements ( $K_D = 10^{-3}$ ) travel further and

Figure 6. Porosity (a,b) and effective pressure (c,d) with decompaction weakening ( $\sigma^b$ ) after  $T=1.5\,\mathrm{Myr}$  for  $\phi^b_0$  (a,c) and  $\phi^c_0$  (b,d)

enrich the upper layer. Furthermore, this enrichment seems to be traveling slightly faster in the region where the initial fluid content was higher (central region of the domain). This requires that the propagation velocity of the enrichment front is not constant and moves further from the location of the porosity discontinuity, which is located exactly at the middle of the domain  $(x_2 = 10 \, \mathrm{km})$ . In contrast, for the case of the negative initial-porosity jump, the enrichment is negligible and is located in the area just above the discontinuity. A marked feature of the discontinuous models shown in Figure 8 is that, exactly at the discontinuity, we observe a significant enrichment or depletion of  $\mathcal C$  depending whether we have a drop  $(\phi_0^{\rm b})$  or an increase  $(\phi_0^{\rm c})$  in the initial porosity.

Finally, Figure 9 shows the resulting normalized tracer element  $\mathcal{C}/\mathcal{C}_0$  for the case of decompaction weakening ( $\sigma^b$ ) and an initially discontinuous porosity  $\phi_0$ . The associated HM model can be found in Figure 6. The resulting cases show marked differences and can be summarized as follows. The case with negative jump discontinuity ( $\phi_0^b$ ) shows a marked enrichment with respect to the incompatible element. In particular, there is a marked enrichment at the discontinuity (at  $x_2 = 10 \,\mathrm{km}$ ), and within the channel in general. Interestingly, for the case of positive jump discontinuity ( $\phi_0^c$ ), the enrichment of the incompatible

Figure 7.  $C/C_0$  after  $T=1.5\,\mathrm{Myr}$  with an initially continuous porosity  $(\phi_0^\mathrm{a})$  without decompaction weakening  $(\sigma^\mathrm{a})$  (a) and with decompaction weakening  $(\sigma^\mathrm{b})$  (b)

**Figure 8.**  $\mathcal{C}/\mathcal{C}_0$  without decompaction weakening  $(\sigma^a)$  after  $T=1.5\,\mathrm{Myr}$  for  $\phi_0^b$  (a) and  $\phi_0^c$  (b)

element is localized close to the discontinuity location (but is smaller at the discontinuity itself). This is explained by the fact that the fluid spreads beyond this point as it was shown in Figure 6 (b,d).

# 5 Discussion and Conclusions

We have presented results for the case of compaction-driven fluid flow in relation to fluid migration in the deep subsurface. Our method aims to resolve the effects of discontinuous porosity distributions as already discussed in Bachmayr and Boisserée (2025). The models confirm previous findings for the cases of homogeneous initial porosity ( $\phi_0$ ) distribution Räss et al. (2018);

Figure 9.  $C/C_0$  with decompaction weakening  $(\sigma^b)$  after  $T=1.5\,\mathrm{Myr}$  for  $\phi_0^b$  (a) and  $\phi_0^c$  (b)

**Figure 10.** Cross section of Figure 9 for  $x_1 = 5 \,\mathrm{km}$  with  $\mathcal{C}/\mathcal{C}_0$  on the y-axis

Connolly and Podladchikov (1998, 2007); Yarushina et al. (2015). However, for the cases when the  $\phi_0$  has jump discontinuities, our method predicts discontinuous solutions without artificial smoothing due to numerical diffusion. Such results are useful for cases where the mechanical variables, such as the effective and fluid pressure, need to be quantified in applications Räss et al. (2018); Yarushina et al. (2022), and thus, our approach can be used to provide a reference case for numerical benchmarks.

An additional advantage of the space-time method is that the one-way coupling of the HM problem to the CT problem can be easily solved using the pre-calculated results of the HM problem. This allows for the investigation of the behavior of various trace elements and the overall mass transport in rock formations that have discontinuous porosity. Our results confirm previous data which suggest that incompatible elements are the most mobile and can travel together with the fluid Richter (1986). This selective enrichment in incompatible elements becomes more prominent in cases where the flow is channelized, leading to the formation of localized geochemical and mineralogical anomalies. Although channeling mechanisms have been discussed in previous works Aharonov et al. (1997); Spiegelman and Kelemen (2003); Schiemenz et al. (2011), the mechanism for the

channeling in our case is different. In the aforementioned studies, the formation of channels was due to the selective dissolution of matrix minerals Schiemenz et al. (2011); Spiegelman et al. (2001). In contrast, in our case the channeling is the result of decompaction weakening Connolly and Podladchikov (2007); Yarushina et al. (2015, 2020). In any case, it becomes apparent that, whatever the localization mechanism may be, the localization of the fluid amplify the enrichment of incompatible elements significantly. Furthermore, our new results also show the the interaction of a fluid-filled channel with a jump discontinuity in the initial porosity. This example is very relevant for the case of fluid transport across heterogeneous layers. In particular, the results show a marked enrichment of the incompatible trace elements at the initial porosity discontinuity for the cases where the initial porosity exhibits a negative jump (i.e. porosity drops sharply at the transition). In the case of a positive jump, we observe a marked depletion at exactly the same location. The results indicate that both porosity and the incompatible-element enrichment, that are associated to the discontinuity, do not move over time and remain at the same location.

Our results indicate that the effects of the channeling of the flow together with the presence of initial discontinuities will produce a variety of element-enrichment patterns that can be investigated in future studies that focus on particular element behavior. These results can be very important in targeted mineral exploration and in the understanding of ore-formation processes.

*Code availability.* All the Julia scripts and data necessary to reproduce the results and figures of this contribution are provided in the Zenodo repository Boisserée et al. (2025) (https://doi.org/10.5281/zenodo.13986982).

# Appendix A: Derivation of the chemical model

Starting with the conservation of mass in the two phases, we get

$$\partial_t(\phi \rho^f \chi_i^f) + \operatorname{div}_x(\phi \rho^f \chi_i^f \mathbf{v}_i^f) = \Gamma_i^f, \tag{A1a}$$

$$\partial_t ((1-\phi)\rho^{\mathbf{s}}\chi_i^{\mathbf{s}}) + \operatorname{div}_x ((1-\phi)\rho^{\mathbf{s}}\chi_i^{\mathbf{s}}\mathbf{v}_i^{\mathbf{s}}) = \Gamma_i^{\mathbf{s}}, \tag{A1b}$$

for  $i=1,\ldots,n$  chemical elements, where  $\Gamma_i^{\rm f}$ ,  $\Gamma_i^{\rm s}$  denote reaction terms. Note that  $\sum_{i=1}^n \chi^{\rm f} = \sum_{i=1}^n \chi^{\rm s} = 1$  and  $\Gamma_i^{\rm f} + \Gamma_i^{\rm s} = 0$  for all  $i=1,\ldots,n$ , since chemical elements can only be exchanged between the two phases. Defining the barycentric velocities

$$\mathbf{v}^{\mathbf{f}} = \sum_{i=1}^{n} \chi_{i}^{\mathbf{f}} \mathbf{v}_{i}^{\mathbf{f}}, \quad \mathbf{v}^{\mathbf{s}} = \sum_{i=1}^{n} \chi_{i}^{\mathbf{s}} \mathbf{v}_{i}^{\mathbf{s}}, \tag{A2}$$

we rewrite (A1)

$$\partial_t \left( \phi \rho^{\mathrm{f}} \chi_i^{\mathrm{f}} \right) + \mathrm{div}_x \left( \phi \rho^{\mathrm{f}} \chi_i^{\mathrm{f}} \mathbf{v}^{\mathrm{f}} \right) + \mathrm{div}_x \left( \phi \rho^{\mathrm{f}} \chi_i^{\mathrm{f}} (\mathbf{v}_i^{\mathrm{f}} - \mathbf{v}^{\mathrm{f}}) \right) = \Gamma_i^{\mathrm{f}}, \tag{A3a}$$

$$\partial_t \left( (1 - \phi) \rho^{\mathbf{s}} \chi_i^{\mathbf{s}} \right) + \operatorname{div}_x \left( (1 - \phi) \rho^{\mathbf{s}} \chi_i^{\mathbf{s}} \mathbf{v}^{\mathbf{s}} \right) + \operatorname{div}_x \left( (1 - \phi) \rho^{\mathbf{s}} \chi_i^{\mathbf{s}} (\mathbf{v}_i^{\mathbf{s}} - \mathbf{v}^{\mathbf{s}}) \right) = \Gamma_i^{\mathbf{s}}, \tag{A3b}$$

and use that for trace elements the diffusion fluxes obey the Fickean limit

$$\rho^{f} \chi_{i}^{f} (\mathbf{v}_{i}^{f} - \mathbf{v}^{f}) = -D_{i}^{f} \nabla_{x} (\rho^{f} \chi_{i}^{f}), \tag{A4a}$$

$$\rho^{s} \chi_{i}^{s} (\mathbf{v}_{i}^{s} - \mathbf{v}^{s}) = -D_{i}^{s} \nabla_{x} (\rho^{s} \chi_{i}^{s})$$
 (A4b)

for both the fluid and solid phase. For advection dominated problems, the diffusion coefficients  $D_i$  are very small and hence we can cancel the corresponding terms in (A3). Adding the resulting equations yields

$$\partial_t \left( \phi \rho^{\mathrm{f}} \chi_i^{\mathrm{f}} + (1 - \phi) \rho^{\mathrm{s}} \chi_i^{\mathrm{s}} \right) + \operatorname{div}_x \left( \phi \rho^{\mathrm{f}} \chi_i^{\mathrm{f}} \mathbf{v}^{\mathrm{f}} + (1 - \phi) \rho^{\mathrm{s}} \chi_i^{\mathrm{s}} \mathbf{v}^{\mathrm{s}} \right) = \Gamma_i^{\mathrm{f}} + \Gamma_i^{\mathrm{s}} = 0. \tag{A5}$$

Next, we define the concentration ratio  $K_D^i = \frac{\rho^s \chi_i^s}{\rho^f \chi_i^f}$  as well as the total concentration  $C_i = (\phi + (1 - \phi)K_D^i)\rho^f \chi_i^f$  which allows us to rewrite (A5) further as

$$\partial_t \mathcal{C}_i + \operatorname{div}_x(\mathcal{C}_i \mathbf{v}_i^{\mathrm{e}}) = 0$$
 (A6)

where  $\mathbf{v}_i^{\rm e} = \frac{\phi \mathbf{v}^{\rm f} + (1-\phi)\mathbf{v}^{\rm s}}{\phi + (1-\phi)K_D^i}$ . Note that this equation is equivalent to (3) where we omitted the index i for convenience and assumed that  $\mathbf{v}^{\rm s} \approx 0$ .

# Appendix B: Handling discontinuous velocities

In order to solve (14) for a discontinuous velocity field ve, we need to ensure mass balance at the discontinuity. This is normally done via the Rankine-Hugoniot jump condition (see, for example, Anderson, 1990, Sec. 4.3 or LeVeque, 2002, Sec. 11.8), which in this case (3) leads to

$$c_{+}\mathbf{v}_{+}^{\mathbf{e}} \cdot \boldsymbol{n} = c_{-}\mathbf{v}_{-}^{\mathbf{e}} \cdot \boldsymbol{n} \tag{B1}$$

where  $c_+, c_-, \mathbf{v}_+^{\text{e}}, \mathbf{v}_-^{\text{e}}$  denote the values of c and  $\mathbf{v}^{\text{e}}$  on both sides of the discontinuity and n is the normal vector with respect to the discontinuity. In the test cases shown in Figures 8 and 9 we have  $n = e_2$ , which simplifies the numerical calculations.

In practice, when running the explicit Euler code to solve (14), we ensure that each time step does not advect the total concentration across the discontinuity. Once the total concentration reaches the discontinuity, we recalculate the mass flux and use it to evaluate the concentration jump (without loss of generality we call it  $c_{-}$ ), as follows

$$c_{+} = c_{-} \frac{\mathbf{v}_{-}^{\mathbf{e}} \cdot \mathbf{n}}{\mathbf{v}_{-}^{\mathbf{e}} \cdot \mathbf{n}}.$$
(B2)

# 335 Appendix C: Comparison with finite difference code

Here we compare our space-time approach with a classical finite difference scheme. Note that for simplicity we chose a onedimensional test case without decompaction weakening. Hence, this is similar to the tests shown in Figure 3 (a,c) and 4. In Figure C1 one can see the error of the finite difference scheme at the terminal time; note the very different rates for the

**Figure C1.** Errors of a finite difference approximation of the one-dimensional hydromechanical model without decompaction weakening for porosity (a) and effective pressure (b)

discontinuous and continuous test cases. This difference is more pronounced for the porosity in Figure C1 (a) since the correct solution is discontinuous whereas the corresponding effective pressure is still continuous with a kink at the location of the discontinuity. Hence, the convergence rates for the pressure in Figure C1 (b) are higher than for the porosity, even though they are still lower than in the case of a continuous initial porosity. Note also that the rates in the continuous case are the theoretically optimal ones. In comparison, the space-time approach does not suffer from slow rates in discontinuous cases as one can see in Figure C2. In addition, the rates here are optimal as well. However, we would like to emphasize that a direct comparison is not possible. This is because the space-time approach is fundamentally different from the finite difference one. For the finite differences, we measure the  $L_2(\Omega)$ -error at the terminal time, whereas for the space-time approach we considered a more complicated space-time error norm. In addition, the number of degrees of freedom (dofs) is not directly comparable since in Figure C1 they correspond to the number of dofs at the terminal time (even though before there were many time steps involved) and in Figure C2 they correspond to the total number of dofs for the entire space-time grid. Finally, we note that the norms measuring the errors of the porosity in Figure C2 (a) and of the effective pressure in Figure C2 (b) are of different kind. In summary, even though the convergence rates of the two methods are not directly comparable, the new space-time approach does not suffer from reduced convergence rates in the presence of discontinuous initial porosities.

**Figure C2.** Space-time errors of the one-dimensional hydromechanical model without decompaction weakening for porosity (a) and effective pressure (b)

# Appendix D: Continuous approximation

In this section we want to compare the approximation of the hydromechanical and chemical model for the discontinuous initial function  $\phi_0^b$  and a continuous approximation of it. In Figure D1, we plot a cross section of both the discontinuous initial function and its smooth approximation.

Figure D1. Cross section of initial porosity  $\phi_0^{\rm b}$  (a) and its smooth approximation (b) for  $x_1=5\,{\rm km}$ 

In Figure D2 (a,c), one can see the hydromechanical model solution to discontinuous case (as in Figure 6 (a,c)) whereas in Figure D2 (b,d) the results were obtained using the continuous (even though very steep) initial function shown in Figure D1 (b). Here, one can see no major difference between the two approaches. This shows that the solution of the continuous approxima-

Figure D2. Porosity (a,b) and effective pressure (c,d) with decompaction weakening after  $T = 1.5 \,\mathrm{Myr}$  for  $\phi_0^\mathrm{b}$  (a,c) and its continuous approximation (b,d)

tion indeed approximates the discontinuous one if the continuous function is steep enough. However, as it is shown in the cross section in Figure D3, the continuous approach still misses most of the steep gradient of  $\phi$  at the position of the discontinuity. This can be improved by using an even steeper approximation of  $\phi_0^b$ , which, on the other side, increases the computational complexity and slows the computation times.

A very similar behavior can be observed when looking at the chemical enrichment patterns connected to the hydromechanical model results. The chemical enrichment patterns are shown in Figure D4 and the corresponding hydromechanical models are shown in Figure D2. Here, the smooth approximation results in a slightly blurred version compared to the discontinuous problem. Only when looking at the cross section in Figure D5, we see a considerable difference of the enrichment at the

**Figure D3.** Cross section of Figure D2 for  $x_1 = 5 \,\mathrm{km}$  with porosity (a,b) and effective pressure (c,d)

Figure D4.  $\mathcal{C}/\mathcal{C}_0$  with decompaction weakening  $(\sigma^b)$  after  $T=1.5\,\mathrm{Myr}$  for  $\phi_0^b$  (a) and its continuous approximation (b)

discontinuity (at  $x_2 = 10 \,\mathrm{km}$ ) which can, for example, have a significant impact on the creation of ore deposits at specific layers in the subsurface.

**Figure D5.** Cross section of Figure D4 for  $x_1 = 5 \,\mathrm{km}$  with  $\mathcal{C}/\mathcal{C}_0$  on the y-axis. (a) corresponds to the discontinuous solution whereas (b) corresponds to the continuous approximation

*Author contributions*. All three authors developed the ideas, S.B. and M.B. developed the hydromechanical solver and all coauthors developed the chemical tracer transport solver on the space-time domain. All the implementations were carried out by S.B.. The manuscript was written and edited by all three authors.

Competing interests. The authors declare that they have no competing interests.

Acknowledgements. The authors would like to thank the reviewers for comments that allowed us to improve the manuscript. S.B. has been funded in part by the German Research Foundation (DFG) – project number 442047500 – SFB 1481. E.M. would like to acknowledge the German Research Foundation (DFG) – project number 521637679 for financial support and Prof. Y. Podladchikov for discussions regarding the discontinuous solutions. M.B. acknowledges support by the German Research Foundation (DFG) – project number 442047500 – SFB 1481.

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
