# Peer review of "Fluid flow channeling and mass transport with discontinuous porosity distribution"

_EGUsphere, 2024_

## Referee Comment (RC1)

This manuscript explores fluid flow and mass transport through a discontinuous porosity distribution, a common feature in layered geological formations. The authors present numerical results obtained using a novel space-time method capable of handling these discontinuities, unlike traditional methods that often smooth them out. Examining the results, this method appears promising. However, improvements are needed to adequately justify the advantage of applying this new method to study fluid flow through a discontinuous porosity. A major concern is that the present work is an application of a new method that was initially presented in a non-peer-reviewed work (an arXiv manuscript). This could significantly impact the reliability of this work. Therefore, I suggest a major revision to include as much content as possible in the theory section to convince readers that this new method is reliable without needing to consult an unpublished work. Please also see the specific comments below.

- The abstract needs to be clearer about the key contribution of this manuscript. Is it the 'novel space-time method' itself? Or was this method developed elsewhere, and this work applies it to study porous flow through a discontinuous boundary? If it is the former, it would be better to briefly explain the novelty in the abstract. If it is the latter, it needs to briefly report more details of the results. For example, what is the "influence" of layering? How does this method help to distinguish such influence, which could be missed by other methods?

- P2. "In addition, we utilize a newly developed space-time method that has been shown to be more accurate in solving this particular problem in the presence of jump discontinuities [1]." This new method is an essential element of this work; however, it refers to a non-peer-reviewed citation. While this could significantly impact the credibility of the results, the authors might consider including results demonstrating that this method is indeed 'more accurate' for this specific problem. I could not find such content in the results section.

- P2. "An additional advantage of the space-time method is that it can be coupled to simple models of chemical-tracer transport in a straightforward manner." I do not understand why this space-time method makes the coupling more straightforward than in other methods using a smoothing scheme.

- Eq 1a and 1b need a reference or a more detailed explanation of how they are derived from the standard mass and momentum equations.

- P2, "...$\frac{\eta_b \sigma}{(1-\phi)\phi^m}$ can be regarded as effective viscosity..." is confusing as there is no such coefficient in Eq 1.

- P3, regarding the equations for $v^e$ and $v^f$: Firstly, please number these equations (and other equations onwards that have not been numbered). Secondly, are both equations only valid when the solid velocity is zero? Is this a assumption applied in the entire work? Why assuming this then? What is the general formulation without such a limitation?

- P3, why can the concentration ratio KD be assumed to be a constant?

- P4, Eq 3 and Eq 1 are inconsistent. In Eq 3, within the brackets on the RHS, the first term of 3a is the same as the last term of 3b. However, the corresponding terms in Eq 1a and 1b are different.
- P4, the meaning of $\bar{p}$, $\varphi$, $\text{div}_x$, $p(0, \cdot)$, and $p_0$ is unclear. Please explain them.
- P4, It is not immediately clear how Eq 4 becomes Eq 5. Please explain this more clearly, possibly with some intermediate steps.
- P4, "The resulting adaptive scheme..." It would be greatly helpful to explain what you mean by 'adaptive' here and how this scheme is adaptive in the calculation. Figure 1 is shown as an example to illustrate the space-time grids. However, some essential details are missing to understand the figure, for example, the meaning of the colors in panel (b).
- P4, It is not clear why the discontinuity of porosity does not cause a problem here. The mathematical formulation still contains the gradient of porosity, which yields a singular value where the discontinuity is prescribed. How has this issue been resolved in this new scheme?
- P5, sec 2.3. The non-dimensionalised values seem to have mistakes. Based on the definitions, the non-dimensionalized values are actually Q=1/600 and T=23.65. Please check these, and also the following calculations if these values are incorrect.
- P6 onwards. The results show that the location of the discontinuity does not vary over time. It seems to remain as a straight line throughout. Why does this discontinuous boundary not advect with the solid phase?

- The results in Section 3 need a discussion to justify the advantage of this new method: what novel feature can only this new method resolve that a conventional method using a sharp but smooth transition can not resolve? How important is this new feature in understanding the actual physical world?

References:

[1] M. Bachmayr and S. Boisseree. An adaptive space-time method for nonlinear poroviscoelastic flows with discontinuous porosities, 2024.

---

## Author Comment (AC1)

**Response to Reviewers' comments on the manuscript**
**"Fluid flow channeling and mass transport with discontinuous porosity distribution"**

by Simon Boisserée, Evangelos Moulas and Markus Bachmayr

We thank the referees for the valuable comments and suggestions that helped to significantly improve the quality of the paper. In fact, their criticism allowed us to perform additional models and tests to justify our points of view. Our answers (in blue) to the reviewers' questions can be found below. For convenience we include the reports made by the reviewers. Our manuscript changes are highlighted in blue in the new version of the manuscript.

**Reviewer 1**

This manuscript explores fluid flow and mass transport through a discontinuous porosity distribution, a common feature in layered geological formations. The authors present numerical results obtained using a novel space-time method capable of handling these discontinuities, unlike traditional methods that often smooth them out. Examining the results, this method appears promising. However, improvements are needed to adequately justify the advantage of applying this new method to study fluid flow through a discontinuous porosity. A major concern is that the present work is an application of a new method that was initially presented in a non-peer-reviewed work (an arXiv manuscript). This could significantly impact the reliability of this work. Therefore, I suggest a major revision to include as much content as possible in the theory section to convince readers that this new method is reliable without needing to consult an unpublished work.

**Reply.** We thank the reviewer for the comment. Regarding the new method: we accept the point of view of the reviewer. The reason why we cited an arxiv was because this manuscript was still under revision (at another journal) and, at the same time, we did not want to duplicate the same material in this manuscript. We have now confirmation of the acceptance of the technical manuscript (that describes in detail the space-time method – citation [3]) and the interested reader can follow that part. Regarding the novelty in this work: we have combined for the first time the new hydromechanical space-time solver with a model of elemental transport. As discussed in the manuscript, this way of coupling offers a significant advantage since the hydromechanical model needs to be computed only once, and the chemical transport model can be viewed as a post-processing step. The new results show that, for the case of incompatible elements, there is a marked enrichment at the initial porosity discontinuity that can be relevant for studies of metasomatism and ore formation. Note that in this version of the manuscript we added a comparison between a continuous approximation and the discontinuous method that clearly shows that the magnitude of the enrichment can be underestimated if the discontinuity is not resolved (Appendix D). In addition, there can be a significant convergence improvement using our method (Appendix C). In the new version, we have tried to emphasize these results (problem solutions and convergence analysis) in order to justify the use of this new method.

Please also see the specific comments below.

1. The abstract needs to be clearer about the key contribution of this manuscript. Is it the 'novel space-time method' itself? Or was this method developed elsewhere, and this work applies it to

study porous flow through a discontinuous boundary? If it is the former, it would be better to briefly explain the novelty in the abstract. If it is the latter, it needs to briefly report more details of the results. For example, what is the "influence" of layering? How does this method help to distinguish such influence, which could be missed by other methods?

**Reply.** We have modified the abstract and we now emphasize on the novel results that are related to the enrichment at the discontinuous interface.

2. P2. "In addition, we utilize a newly developed space-time method that has been shown to be more accurate in solving this particular problem in the presence of jump discontinuities [1]." This new method is an essential element of this work; however, it refers to a non-peer-reviewed citation. While this could significantly impact the credibility of the results, the authors might consider including results demonstrating that this method is indeed 'more accurate' for this specific problem. I could not find such content in the results section.

**Reply.** We have added a comparison with a finite difference scheme in Appendix C. The results in this appendix show the performance difference of the two methods. In addition, we show (in the new Appendix D) the differences of a continuous code that is tuned to "capture" the effects near the discontinuity. We explain the differences in a more quantitative manner in the new version of the manuscript. The difference both in the performance and in the results justify the use of such a method when strong gradients in porosity are modelled.

3. P2. "An additional advantage of the space-time method is that it can be coupled to simple models of chemical-tracer transport in a straightforward manner." I do not understand why this space-time method makes the coupling more straightforward than in other methods using a smoothing scheme.

**Reply.** The coupling is simple because it can be carried out as a post-processing step. This is because the chemical tracers do not affect the mechanical properties of the system (but the fluid velocity affects the movement of the trace element). Thus, by solving the hydromechanical model once, allows the storage of the space-time solution (the adaptive grid allows very efficient storage). Then, having the solution in space and time, the chemical tracer problem can be solved with the method of characteristics, that is highly parallelizable. We have modified the text to address this issue.

4. Eq 1a and 1b need a reference or a more detailed explanation of how they are derived from the standard mass and momentum equations.

**Reply.** We added two references for this model directly after eq. (1).

5. P2, "... $\frac{\eta_b \sigma}{(1-\phi)\phi^m}$ can be regarded as effective viscosity..." is confusing as there is no such coefficient in Eq 1.

**Reply.** We removed this part.

6. P3, regarding the equations for $\mathbf{v}^e$ and $\mathbf{v}^f$: Firstly, please number these equations (and other equations onwards that have not been numbered). Secondly, are both equations only valid when the solid velocity is zero? Is this a assumption applied in the entire work? Why assuming this then? What is the general formulation without such a limitation?

**Reply.** All equations are numbered now. We have modified the text and added a new section (1.2) that addresses and justifies our approximations.

7. P3, why can the concentration ratio $K_D$ be assumed to be a constant?

   **Reply.** We have modified the text and explain in detail the factors that affect the partition coefficient. In principle it can vary, but in cases where temperature changes are small (i.e. adiabatic cooling or close to the edge of a mantle wedge) $K_D$ can vary weakly with pressure (composition is assumed constant). In this work, we have considered a constant $K_D$ for simplicity. Furthermore, even if an incompatible element that has a $K_D$ of 0.001 becomes 10 time larger, will not affect the general conclusions of this study. In principle, we could have more case-specific models that would account for the change in $K_D$. However, we find that a total systematics of this variation goes beyond the scope of this work.

8. P4, Eq 3 and Eq 1 are inconsistent. In Eq 3, within the brackets on the RHS, the first term of 3a is the same as the last term of 3b. However, the corresponding terms in Eq 1a and 1b are different.

   **Reply.** We thank the reviewer for pointing it out. We have now added a missing $\eta_b$ in Eq 1.

9. P4, the meaning of $\bar{p}$, $\varphi$, $\mathrm{div}_x$, $p(0, \cdot)$ and $p_0$ is unclear. Please explain them.

   **Reply.** We rewrote (9) to eliminate $\bar{p}$, the definition of $\varphi$ is highlighted in (5). We defined $\mathrm{div}_x$, $\nabla_x$ and the notation $g(0, \cdot)$ after (1). $p_0$ was already defined there, and we added it in (6) and (8) to make it more clear.

10. P4, It is not immediately clear how Eq 4 becomes Eq 5. Please explain this more clearly, possibly with some intermediate steps.

    **Reply.** We added an explanation below (10).

11. P4, "The resulting adaptive scheme..." It would be greatly helpful to explain what you mean by 'adaptive' here and how this scheme is adaptive in the calculation. Figure 1 is shown as an example to illustrate the space-time grids. However, some essential details are missing to understand the figure, for example, the meaning of the colors in panel (b).

    **Reply.** The scheme is adaptive because it yields reliable local error estimators which can be used to steer an adaptive grid refinement routine. We extended the manuscript to make this more clear. Furthermore we added a description of the colors in the caption of Fig. 1.

12. P4, It is not clear why the discontinuity of porosity does not cause a problem here. The mathematical formulation still contains the gradient of porosity, which yields a singular value where the discontinuity is prescribed. How has this issue been resolved in this new scheme?

    **Reply.** We have looked into this issue and have expanded our analysis by adding a new section (Appendix B) regarding the treatment of jump discontinuities. In the new version, every time there is a jump discontinuity, we proceed with the Rankine-Hugoniot jump conditions at the interface. This ensures mass conservation even in the pure discontinuous case.

13. P5, sec 2.3. The non-dimensionalised values seem to have mistakes. Based on the definitions, the non-dimensionalized values are actually Q=1/600 and T=23.65. Please check these, and also the following calculations if these values are incorrect.

    **Reply.** We added more details about the nondimensionalization at the beginning of Section 2.3. We realized that the issue was the definition of Deborah number following previous literature. The problem was that, in previous literature, the lengthscale in the definition of the Deborah number was

not independent and was depending on the definition of the compaction lenghtscale (which in turn depends on the viscosity). In our approach, we chose a fixed lengthscale relevant to our geological problem. In this case, we do not have a "Deborah" number as it was previously stated. We have modified the text explaining our assumptions in more detail (we have removed any reference to the Deborah number as well). In summary, our scaling is correct, but it is different compared to the previous literature.

14. P6 onwards. The results show that the location of the discontinuity does not vary over time. It seems to remain as a straight line throughout. Why does this discontinuous boundary not advect with the solid phase?

    **Reply.** It appears that since $\mathbf{v}^s = 0$ and $\phi$ is property of solid, it does not move (the waves move through it). We added a paragraph about this in Sec. 3. More importantly, this agrees with the theoretical results in [2, Thm. 4.6].

15. The results in Section 3 need a discussion to justify the advantage of this new method: what novel feature can only this new method resolve that a conventional method using a sharp but smooth transition can not resolve? How important is this new feature in understanding the actual physical world?

    **Reply.** Apart from the efficient storage of the HM model explained earlier, in the new version of the manuscript, we added a comparison with a finite difference scheme in Appendix C, where we now show that the new method does not suffer from discontinuities in $\phi$ whereas the finite difference approach does.

**Reviewer 2**

This manuscript presents results of numerical simulations of localized porous flow in layered heterogenous media with a jump-discontinuity in the initial porosity and permeability fields. The authors employ a novel numerical method to solve the posed problem, and they show interesting results of a few simulations. The second part of the manuscript shows a potential use case of the results presented in the first part, for calculated element transport and partitioning. I find the topic interesting and relevant for the potential readers of GMD.

Unfortunately, it is not clear for me what big-picture message the manuscript aims to convey. The manuscript superficially touches on several key points, but in my opinion the depth of the discussion is not sufficient to really underpin these key points.

- Is this an article aiming to demonstrate the applicability and usefulness of the adaptive space-time method compared to other, traditional numerical methods?

- Is this an article which investigates the physics of jump discontinuities for localized fluid flow?

- Is this an article aiming to tackle geological problems related to fluid flow and element transport?

One of the key issues I found is that the authors use a novel numerical method, referring to submitted but not accepted manuscripts, without demonstrating the applicability of the method. Moreover, due to the small number of simulations presented and the extremely narrow parameter space explored, it is hard to judge the generality of the results. This would be acceptable if the authors would specifically

target a given geological problem, but the geological description is very general while some of the chosen parameter values are far from typical (i.e. 0.1% background porosity). Finally, the figures have room for improvement (e.g. reducing empty spaces, balancing font size, positioning labels more intuitively).

To summarize, the work presented is worth publishing, but the big-picture message needs further maturation, and the presentation style needs further polishing. Therefore, I recommend accepting the manuscript after major revision.

**Reply.** We thank the reviewer for the constructive criticism. We have now expanded both the abstract and the main text in order to show our message more clearly. Our results show various patterns that emerge when a channel interacts with a discontinuity. This is not an artifact and is supported also by results that attempt to capture the discontinuity using a continuous but a very sharp gradient (see Appendix D in the new manuscript). Regarding the applicability of our choices we added a paragraph justifying our assumptions and choice of parameters (section 1.2). We provide more details below. Overall, since this journal is about model development, we tried to keep a balance between presenting technical details of this method and how this method can be used efficiently in future studies. We hope that the reviewer finds the current version clearer.

Some specific remarks:

1. Please include a benchmark figure comparing the result obtained by the novel method and with at least one traditional method for homogenous and discontinuous background porosity.

   **Reply.** This part was answered, please see 1st reviewer, point 15

2. Please demonstrate the suitability of the novel space-time method by validating it with a discontinuous analytical solution. I understand that such a solution might not exist for porous fluid flow channeling, but there are many other physical processes with known discontinuous solutions.

   **Reply.** We have added a section that shows the convergence of the continuous approach to the discontinuous case at the end (Appendix D). With increasing resolution the results agree (to first order). However, the accurate resolution of the discontinuity becomes computationally inefficient with larger resolutions (see convergence plots for details - Appendix C).

3. Reading the manuscript, I have the impression that the authors consider their novel method superior to all traditional methods because it can handle discontinuities. This is an obvious advantage compared to (non-adaptive) staggered grid finite difference solvers. Nevertheless, it would be instructive to see how much numerical diffusion affects the results. It is not clear to me what is the advantage of the space-time method compared to finite element or finite volume methods with fitted meshes? Those methods were designed for and are regularly used for problems with discontinuous material properties.

   **Reply.** In 1d a finite volume scheme corresponds to a finite difference scheme taking averages, we added a comparison with a finite difference scheme in Appendix C.

4. Please describe the boundary conditions used for the simulations. It seems like that in Figures 4-9 the anomalies either touch the boundaries or get very close to them. Please demonstrate that boundary effects are not distorting the results.

   **Reply.** We ran the hydro-mechanical simulations on a larger domain than depicted and added a paragraph at the beginning of Sec. 3 describing this. The chemical simulations do not encounter problems here, since they are particle-based and do not include any boundary conditions.

5. Maybe choosing absolute porosity and pressure values is not the best way to show the results. I would include subplots showing porosity and pressure change compared to the initial values with a diverging colormap centered at zero.

   **Reply.** Since the effective pressure $p$ can be negative and especially 0 (e.g. at the initial time we always choose $p_0 = 0$), this cannot be done here. For $\phi$ it is possible but to have a better comparison with the absolute values of $p$ here, we would prefer to keep them absolute as well. Another reason why we prefer to keep our values specific is our choice of parameters. Since we did not perform systematic studies by varying the non-dimensional numbers, we found that presenting the results in dimensional form will be more intuitive for a broader geoscientific audience.

6. I find it confusing that the on many figures colorbars and their corresponding labels are on opposite sides of the figures.

   **Reply.** The labels are now next to the colorbars.

7. Choosing 0.1% background porosity with 0.1% as the maximum initial perturbation is extremely peculiar. This is two orders of magnitude lower than the typical range for most porous fluid flow applications. Please either demonstrate that the results are applicable for a wide porosity range or specify from the beginning the geological settings and processes for which these values are representative.

   **Reply.** We have now explained our application in more detail. We have added a new section (1.2) justifying our choices.

8. The concentration equation is not just a transport equation. It is a reactive transport equation, as it implicitly forces a constant concentration ratio, implying mass/element transfer between the two phases. Please expand the description of the chemical model, and make sure the element partitioning is mentioned from the beginning.

   **Reply.** We added details regarding the derivation of the chemical equation in Appendix A.

9. Equation 2 seems to imply that $\mathcal{C}$ must be conserved (if this is not the case, please explain it in the manuscript). Comparing the results of Figures 8-9, it seems that the total amount of $\mathcal{C}$ in the domain is different for the two models. Is that a boundary effect? If that is the case, please choose a model configuration where boundary effects are negligible.

   **Reply.** We thank the reviewer for pointing that out. In the previous version of the manuscript we had a small error because of the jump condition. We now have recalculated the results and treat the discontinuity exactly. Details about the new implementation can be found in Appendix B.

References: [1] M. Bachmayr and S. Boisseree. An adaptive space-time method for nonlinear porovis-coelastic flows with discontinuous porosities, 2024.